# Green Extraction of Forsythoside A, Phillyrin and Phillygenol from *Forsythia suspensa* Leaves Using a β-Cyclodextrin-Assisted Method

**DOI:** 10.3390/molecules27207055

**Published:** 2022-10-19

**Authors:** Jing Li, Qiao Qin, Sheng-Hua Zha, Qing-Sheng Zhao, Hang Li, Lu-Peng Liu, Shou-Bu Hou, Bing Zhao

**Affiliations:** 1College of Bioscience and Engineering, Hebei University of Economics and Business, Shijiazhuang 050061, China; 2College of Food and Biology, Hebei University of Science and Technology, Shijiazhuang 050018, China; 3State Key Laboratory of Biochemical Engineering, Institute of Process Engineering, Chinese Academy of Sciences, Beijing 100190, China; 4Beijing Tong Ren Tang Health Pharmaceutical Co., Ltd., Beijing 100085, China

**Keywords:** *Forsythia suspensa* leaves, β-cyclodextrins, response surface methodology, stability, antioxidant activity

## Abstract

In this study, a green process of β-cyclodextrin (β-CD)-assisted extraction of active ingredients from *Forsythia suspensa* leaves was developed. Firstly, the optimal process of extraction was as follows: the ratio between *Forsythia suspensa* leaves and β-CD was 3.61:5, the solid–liquid ratio was 1:36.3, the temperature was 75.25 °C and the pH was 3.94. The yields of forsythoside A, phillyrin and phillygenol were 11.80 ± 0.141%, 5.49 ± 0.078% and 0.319 ± 0.004%, respectively. Then, the structure characteristics of the β-CD-assisted extract of *Forsythia suspensa* leaves (FSE-β-CD) were analyzed using powder X-ray diffraction (PXRD), Fourier transform infrared spectroscopy (FT-IR), differential scanning calorimetry (DSC), scanning electron microscopy (SEM) and molecular docking to demonstrate that the natural active products from *Forsythia suspensa* leaves had significant interactions with the β-CD. Additionally, the loss of forsythoside A from aqueous FSE-CD at 80 °C was only 12%, compared with *Forsythia suspensa* leaf extract (FSE) which decreased by 13%. In addition, the aqueous solubility of FSE-CD was significantly increased to 70.2 g/L. The EC_50_ for scavenging DPPH and ABTS radicals decreased to 28.98 ug/mL and 25.54 ug/mL, respectively. The results showed that the β-CD-assisted extraction process would be a promising technology for bioactive compounds extracted from plants.

## 1. Introduction

*Forsythia suspensa* is a plant widely grown in China, Korea, Japan and many European countries [1]. Its fruit is a traditional Chinese herbal medicine used to treat fever, inflammation, gonorrhea, carbuncle and erysipelas [2]. Although *Forsythia suspensa* fruits are not a food material to be applied to the food industry [3], *Forsythia suspensa* leaves containing similar ingredients have been allowed in recent years by several countries to apply in the food industry. For example, it was approved to be used as food for the first time by the local food safety standard of Shanxi Province, China in 2017. Forsythoside A, phillyrin and phillygenol, considered as the main components in *Forsythia suspense* fruit*,* have been demonstrated to possess a wide variety of biological activities, such as anti-inflammation, anti-tumor, liver protection, anti-virus [4,5] and anti-oxidation [6,7]. It was reported that contents of forsythoside A and phillyrin in *Forsythia suspensa* leaves were much higher than those in *Forsythia suspensa* fruits [1]. Moreover, it was confirmed that active ingredients from *Forsythia suspensa* leaves were low-toxic or non-toxic through acute and sub-chronic oral administration in rodent models [8], which opened many new application fields of forsythoside A, phillyrin and phillygenol in functional foods [9]. 

Conventional extraction methods, such as organic solvent extraction [10] and hot water extraction [11], suffer from low yield and high cost. Even more important, solvents such as chloroform, hexane and ether can cause serious pollution, restricting their use in food. In order to improve the extraction rate, some new methods including supercritical CO_2_ extraction [12], microwave-assisted extraction [13], ultrasonic-assisted extraction [14], ionic liquids (ILs) [15], etc., have been used to extract the natural products of *Forsythia suspense* leaves. However, these methods require high costs and precision equipment, which are not suitable for industrial production. Furthermore, ILs cannot be used in the food industry due to their toxicity. In addition, due to the molecular structures of these three compounds having ester bonds and phenolic hydroxyl groups, they are easy to be oxidized and lose their activity during storage. Forsythoside A faces the problems of poor stability under high temperature and acidic and alkaline conditions [16]. Therefore, to obtain bioactive compounds with high stability and solubility, the conventional craftwork of pharmaceuticals requires six steps which include extraction, spray drying, dissolution, interaction with the cyclodextrin and respray drying, which are highly cumbersome and costly, as demonstrated in Figure 1b [17]. Consequently, it is urgent to explore cost-effective and eco-friendly technology for the extraction of forsythoside A, phillyrin and phillygenol from *Forsythia suspensa* leaves for their development and application in functional food, cosmetic, nutraceutical and pharmaceutical products.

β-CD is a common pharmaceutical excipient with a special structure of a hydrophilic outer surface and a relatively hydrophobic central cavity forming an ultra-capsular inclusion when mixed with various guest molecules so as to improve the solubility, physical stability and biological properties of the guest molecules [18]. Moreover, after entering the body, the loop of β-CD is opened to form a straight-chain oligosaccharide, which participates in the body metabolism with no accumulation effect and non-toxicity. In recent years, β-CD-assisted extraction has been used to extract phytochemicals from natural sources as a green and effective method [19,20]. For example, Jurga Andreja Kazlauskaite et al. [20] examined the influence of several kinds of cyclodextrins on the extraction of isoflavone from Trifolium pratensis L. It was determined that cyclodextrins considerably boosted the isoflavone aglycone’s yield and improved its water solubility. In work by Gastón Ezequiel Maraulo et al. [21], ultrasonication, stirring and a β-CD aqueous solution were combined to extract the active substances from olive pomace, and it was confirmed that the presence of β-CD altered the dehydrated ex-stability, tract’s hygroscopicity and appearance. In addition, Hiba N. Rajha et al. [22] refined the β-CD-assisted extraction procedure of polyphenols from grape shoots. Their results clearly revealed the potential of β-CD to magnify polyphenol extraction from vine shoots and also increase the stability and oxidation resistance of polyphenols. As shown in Figure 1b, the interaction with cyclodextrin and extraction are consolidated into one step; stable active substances can be obtained through three simple steps in β-CD-assisted extraction technology. However, there was no report about β-CD-assisted extraction of bioactive compounds from *Forsythia suspensa* leaves.

In this study, β-CD-assisted extraction of forsythoside A, phillyrin and phillygenol from *Forsythia suspensa* leaves was explored. It was expected to obtain active ingredients of *Forsythia suspensa* leaves with high extraction yields, good stability and high water solubility. The extraction conditions were optimized with single-factor experiments and response surface methodology (RSM). The interaction between β-CD and active ingredients was inspected using powder X-ray diffraction (PXRD), Fourier transform infrared (FT-IR) spectroscopy, differential scanning calorimetry (DSC) and scanning electron microscopy (SEM). The effects of β-CD inclusion on the stability, aqueous solubility and antioxidant activity of forsythoside A, phillyrin and phillygenol were also studied.

## 2. Results and Discussion

### 2.1. Single-Factor Experiments

#### 2.1.1. Effects of the Solid–Liquid Ratio on Natural Products

In the extraction process, there was a balance between the amount of solvent and the Chinese medicinal herb [23]. Figure 2a shows the effect of the R*_S/L_* on the extraction yields of forsythoside A and phillyrin from *Forsythia suspensa* leaves in β-CD-assisted extraction. It could be seen that when the R*_S/L_* was 1:30, the extraction yields of forsythoside A and phillyrin reached their maximum and then decreased with the decline in R*_S/L_*. Phillygenol showed a downward trend. A small R*_S/L_* reduced the probability of interaction between natural products and β-CD, resulting in the decline in the extraction yields of natural products. According to the comprehensive integral judgment, the R*_S/L_* of 1:30 was suitable for the β-CD-assisted extraction of *Forsythia suspensa* leaves. The specific extraction R*_S/L_* was determined in subsequent RSM experiments.

#### 2.1.2. Effects of Extraction Temperature on Natural Products

The effect of extraction temperature on the extraction yields of natural products of *Forsythia suspensa* leaves in β-CD-assisted extraction is shown in Figure 2c. It could be seen that in the range of 25~70 °C, the extraction yields of natural products were positively correlated with the extraction temperature, and reached a maximum at 70 °C (forsythoside A, 101 mg/g; phillyrin, 54.3 mg/g; and phillygenol, 0.33 mg/g), which might have caused the increased diffusion of solvent into the cells. The desorption and solubility of the natural products from the cells increased, and as all the reactions had negative enthalpy energy, β-CD was more likely to form inclusion complexes with the natural products when the temperature increased [24]. It was also seen that the output of natural products showed a downward trend above 70 °C, which was mainly because the *Forsythia suspensa* extract was unstable at high temperature and some pyrolysis occurred [1]. The optimal extraction temperature of *Forsythia suspensa* leaves with pure-water extraction and β-CD-assisted extraction was approximately 70 °C.

#### 2.1.3. Effects of pH on Natural Products

The effect of pH on the extraction yields of natural products from *Forsythia suspensa* leaves in β-CD-assisted extraction is shown in Figure 2e. It could be seen the extraction yields of natural products from *Forsythia suspensa* leaves were significantly affected by pH. When the pH was lower than 3, the yield of forsythoside A and phillyrin increased significantly with the increase in pH, while the yield of phillygenol decreased drastically. When the pH was higher than 3, the yield of forsythoside A decreased precipitously, which was owing to the instability of forsythoside A under weak alkaline conditions [16]. Therefore, the pH of β-CD-assisted extraction of *Forsythia suspensa* leaves was kept at 3. 

#### 2.1.4. Effects of R*_F/β_* on Natural Products

When the extraction time, R*_S/L_,* extraction temperature and pH were 1 h, 1:10, 75 °C and 3, respectively, the effect of R*_F/β_* in β-CD-assisted extraction on the extraction yields of natural products from *Forsythia*
*suspensa* leaves was explored. It can be seen from Figure 2g that the yields of natural products’ extractions increased when R*_F/β_* increased from 5:1 to 5:4, and then reached the highest value, with forsythoside A, 110 mg/g; phillyrin, 52.7 mg/g; and phillygenol, 0.27 mg/g. This was possibly due to the fact that the higher the concentration of β-CD present in the extraction solution, the more β-CD complex was formed, allowing the efficient extraction from samples [24]. After R*_F/β_* increased higher than 5:4, the yields of the natural products decreased due to the dissolution of the native products that was prevented by the increase in the viscosity of the aqueous solution [25].

### 2.2. Response Surface Optimization of Extraction for β-CD-Assisted Extraction

The Box–Behnken design (BBD) used in this work had four factors, including the R*_S/L_*, extraction temperature, pH and R*_F/β_*, with three levels including twenty-nine runs for optimal process conditions. It can be seen from Appendix A that the highest yield at 6.64% was obtained in the operation #22 when the R*_S/L_,* extraction temperature, pH and R*_F/β_* were 1:40, 85, 3 and 4:5, respectively. In addition, analysis of variance (ANOVA) was performed, and regression models were summarized in Appendix A. The *p*-value of 0.0002 and F-value of 7.81 showed that the model prediction was highly significant. The R^2^ value of this model was 0.8865, indicating that more than 89% of the actual values were consistent with the model-predicted values. In addition, the insignificant lack of fit (*p* > 0.05) and values of pure error indicated the good repeatability of the results. The CV of this model was 4.53%, which was within the recommended level of 5%, showed good reliability of the model [26]. The adeq precision (AP) refers to the consistency of a model for describing the process. The AP value of 10.248 exhibited sufficient consistency of the model with the extraction process. Furthermore, the PRESS value of 5.12 was necessary for the well-fitted model. Through multiple regression analysis, the following second-order polynomials could be obtained (Equation (1)).
*Y = 6.42022 + 0.49994X_1_ − 0.18256X_2_ + 0.276704X_3_ − 0.10739X_1_ + 0.24915X_1_X_2_ − 0.072469 × X_1_X_3_ + 0.052535X_1_X_4_ +**0.43017X_2_X_3_ + 0.067554X_2_X_4_ − 0.901034X_3_X_4_ − 0.34706X_1_^2^ + 0.55742X_2_^2^ − 0.53869X_3_^2^ − 0.21180X_4_^2^*(1)

The three-dimensional response surface diagrams in Figure 3 show the influences of variables on Y (comprehensive score). Figure 3a shows the interaction of extraction temperature (X_2_) and R*_S/L_* (X_1_) on Y (comprehensive score). The effect of extraction temperature on yield was higher than that of R*_S/L_*. The yield gradually increased with the increase in R*_S/L_*, and then reached equilibrium. Figure 3b shows the interaction of pH (X_3_) and R*_S/L_* (X_1_) on Y. The interaction between pH and R*_S/L_* had a very significant impact on yield. When pH reached the middle level, a higher Y value was obtained. Figure 3c shows the interaction of R*_F/β_* (X_4_) and R*_S/L_* (X_1_) on Y; R*_F/β_* had no significant effect on yield. When R*_S/L_* was higher, a higher yield was obtained. Figure 3d shows the interaction of pH and extraction temperature on yield. When pH reached the middle, a higher yield was obtained. In contrast, pH had a higher effect on yield than extraction temperature. Figure 3e shows the interaction of R*_F/β_* (X_4_) and extraction temperature (X_2_) on Y; both of them had little influence on yield, and a higher Y value was obtained at the lower level of extraction temperature and middle level of R*_F/β_*. The interaction of R*_F/β_* (X_4_) and pH (X_3_) on Y is shown in Figure 3f, showing pH had a very significant impact on yield, while R*_F/β_* had a much smaller impact. At the medium level of pH, a larger Y value was obtained.

The highest extraction yields of forsythoside A, phillyrin and phillygenol were 11.80 ± 0.141%, 5.49 ± 0.078% and 0.319 ± 0.004%, respectively, when the R*_S/L_,* extraction temperature, pH and R*_F/β_* was 1:36.3, 75.25 °C, 3.94 and 3.61:5, respectively.

The experimental results and three-dimensional response surface of pure-water extraction are shown in Appendix A. The highest extraction yield was 5.53 ± 0.34% with a R*_S/L_* of 1:23.7, extraction temperature of 84.7 °C and pH of 4.06. 

The extraction yields of natural products from *Forsythia suspensa* leaves were compared between β-CD-assisted extraction and pure-water extraction. Using β-CD-assisted extraction, the extraction yields of forsythoside A, phillyrin and phillygenol were 14.16%, 22.47% and 52.77% higher than that using pure-water extraction, respectively. This result was possibly caused by the special structure of external hydrophilicity and internal hydrophobicity of β-CD [27]. Although phillyrin and phillygenol did not dissolve well in water, when β-CD inclusion complexes were formed, the solubility of natural active substances was increased significantly. Moreover, the concentration gradient of active substances in *Forsythia suspensa* leaves and extraction solutions raised markedly due to the formation of inclusion complexes, which, therefore, led to a noticeable improvement in the extraction yields.

### 2.3. Characterization of β-CD Inclusion Complexes

#### 2.3.1. PXRD Analysis

PXRD was used to identify the physical state of guest molecules and characterize inclusion complexes. It can be seen from Figure 4 that FSE showed diffraction characteristic peaks at the angles of 16.7° and 20.19° in 2θ. β-CD had many strong absorption peaks and a typical crystal structure. Inclusion compounds presented a set of reflections that were characteristic of a new crystalline phase formed by the units of the inclusion compound. However, the 1:1 physical mixed product had not only the crystallization peak of β-CD, but also the outline of the common extracted products of *Forsythia suspensa* leaves. It was a simple superposition of β-CD and the common extracted products of *Forsythia suspensa* leaves, which was the same as the results of Wu [28]. Although in the case of the present work, the material was amorphous and no conclusions could be obtained from single inclusion complexes’ powder XRD diffractograms, the difference in inclusion complexes with the 1:1 physical mixed product powder confirmed that the Forsythia *suspensa* leaves’ products had interactions with the cyclodextrin.

#### 2.3.2. FT-IR Spectroscopy Analysis

The FT-IR spectra of FSE, β-CD, FSE-β-CD and1:1 FSE/β-CD PM are shown in Figure 5. FSE showed high-intensity absorption peaks at 1610 cm^−1^ and 1520 cm^−1^ belonging to the stretching vibrations of the benzene ring skeleton (C=C). The significant peaks at 1270 cm^−1^, 1160 cm^−1^ and 1038 cm^−1^ were assigned to the C-O stretching vibration. There was a pair of double substituted benzene ring vibration peaks at 818 cm^−1^. The IR spectrum of β-CD was characterized by the prominent bands at 1630 cm^−1^ corresponding to the C = C stretching vibration, 1420 cm^−1^ assigned to the -CH_2_ bending vibration and 1150 cm^−1^ assigned to the C-H stretching vibration. There was a particularly significant characteristic absorption peak at 1030 cm^−1^ that assigned to the C-O-C stretching vibration.

As can be seen from the figure, the FSE characteristic peaks showed signs of attenuation in FSE-β-CD, such as at 1520 cm^−1^, 1270 cm^−1^ and 818 cm^−1^. Moreover, the characteristic peaks of FSE at 1160 cm^−1^, 1070 cm^−1^ and 1038 cm^−1^ were masked by the characteristic peaks of β-CD. This was because cyclodextrins limited the vibration of the functional groups of the Forsythia leaf extract which proved that there was an interaction between β-CD and the *Forsythia suspensa* leaf extract. The characteristic peaks of FSE and β-CD were observed in the spectra of their physical mixtures with a stacking effect, indicating weak or no interaction between FSE and β-CD during physical mixing [28]. The characteristic summit of FSE diminished or disappeared after the FSE’s interaction with the β-CD. FT-IR spectra indicated a possible interaction between the host and guest molecules, but other methods needed to be carried out to verify the inclusion complex formation.

#### 2.3.3. Thermal Studies

DSC analysis technology was used to study the physical state of the guest molecules [29]. The DSC curves of FSE, β-CD, FSE-CD and FSE/β-CD PM are shown in Figure 6. The DSC curve of FSE was a smooth straight line, indicating that there was no heat absorption or release of FSE, which indicated that FSE was in a disordered state. Β-CD had a wide absorption peak at 58–100 °C, which was due to dehydration [30], and the phase transition resulted in a high endothermic peak at approximately 310 °C. The DSC curve of FSE-CD showed that the wide absorption peak of β-CD disappeared, indicating that some of the water molecules bound to the CDs were replaced by natural product molecules, which indicated that the natural product was successfully contained in the CD’s cavity. Moreover, the phase transition peak of β-CD was shifted to a lower temperature than that caused by the interaction between FSE and β-CD. As indicated by the above phenomenon, *Forsythia*
*suspensa* leaf extract was ascribed to interact with cyclodextrin.

#### 2.3.4. SEM Analysis

The microstructures of samples are shown in Figure 7d_1_–g_2_. FSE was granular with serious aggregation and adhesion, so its shape was irregular. β-CD was flakey or blocky, indicating its crystalline state. The microstructure of 1:1 FSE/β-CD PM had a mixed state of particles and blocks, and the structure did not change, indicating no interaction between the two in the process of physical mixing. The microstructure of FSE-β-CD showed a concave microcapsule structure, indicating the cyclodextrin interaction with the *Forsythia suspensa* extract. A slight depression in the surface was attributed to the increase in temperature firstly in spray drying and then after cooling down rapidly, leaving the traces of wall deformation not affecting its interaction with the cyclodextrin effect [31]. The microcapsules of FSE-β-CD were similar in size without adhesion and rupture, demonstrating the film-forming property of β-CD as a packaging material for *Forsythia suspensa* leaf extract. The SEM image results effectively supported the analysis of the above FT-IR spectra, PXRD patterns and other characterizations.

#### 2.3.5. Molecular Docking

To gain insights into the molecular interactions between forsythoside A, phillyrin and phillygenol, and β-CD in the inclusion complexes, a molecular docking analysis was performed. The results verified that forsythoside A, phillyrin and phillygenol formed an inclusion complex with β-CD at a 1:1 ratio. As shown in Figure 7a, forsythoside A was effectively encapsulated by β-CD, which showed that forsythoside A was fully embedded in the cavity of β-CD. The two phenolic hydroxyl groups of forsythoside A passed through the β-CD ring and formed hydrogen bonds with β-CD. The docking score for the forsythoside A/β-CD IC was −4.05 kcal/mol, and six hydrogen bonds were formed between the host and guest. Hydrogen bonding played an important role in stabilizing the inclusion complex between forsythoside A and β-CD. This was possibly the reason why the stability of forsythoside A from FSE-β-CD was improved. As shown in Figure 7b,c, both phillyrin and phillygenol were effectively encapsulated by β-CD in a similar interaction pose, which showed that the hydrophobic part of phillyrin and phillygenol was fully embedded in the cavity of β-CD. Glycosides of phillyrin and hydroxyls of phillygenol passed through the β-CD ring and formed hydrogen bonds with β-CD. The docking scores for the phillyrin/β-CD IC and phillygenol/β-CD IC were −5.24 kcal/mol and −5.72 kcal/mol, and two and one hydrogen bond were formed between the host and guest, respectively. Phillyrin and phillygenol’s hydrophobic groups were encapsulated by cyclodextrin, which also explained the reason why the phillyrin and phillygenol extraction yields of the CD-assisted extraction were enhanced, and the water solubility of FSE-β-CD was greatly improved.

### 2.4. Thermal Stability Study

The thermal stabilities of the active ingredients are shown in Figure 8. Forsythoside A in the FSE solution showed a sharp decrease, and phillyrin showed a slower decrease with time. However, forsythoside A and phillyrin degraded more slowly in FSE-β-CD than FSE. Compared with FSE, the content loss of forsythoside A in FSE-CD decreased by 13%. This was possibly due to the interaction between bioactive components and β-CD which made their structure more stable. In addition, the heating process could promote the interaction with the cyclodextrin because the reaction was endothermic. It also could be seen that there was little change in the concentration of phillygenol. This showed that phillygenol had good stability. Therefore, β-CD-assisted extraction could enhance the stability of the natural active substances [17].

### 2.5. Aqueous Solubility Experiments

Solubility of FSE-β can be seen in Figure 8d–e. When FSE-β-CD and FSE were supersaturated in aqueous solution at room temperature, the solubility of FSE-β-CD was 70.2 g/L, while that of FSE was only 6.08 g/L. The concentrations of forsythoside A, phillyrin and phillygenol in the FSE-β-CD solution reached 5.41 g/L, 2.50 g/L and 0.23 g/L, which were increased by 294.25%, 422.18% and 2303.59% respectively. As mentioned above, the outer surface of β-CD was hydrophilic while the inner cavity which could accommodate the *Forsythia suspensa* leaves’ active substances was hydrophobic. Hydrophobic active substances such as phillyrin and phillygenol could interact with the β-CD cavity through intermolecular hydrogen bonds and the van der Waals force. For the exterior hydrophilic surface of CDs, inclusion complexes could have a good wettability, which improved the water solubility of the poorly soluble drugs. This result was consistent with the results of inclusion complexes of cannabidiol with β-cyclodextrin [32].

### 2.6. Antioxidant Activity Study

The antioxidant properties of the inclusion complexes of natural products from *Forsythia suspensa* leaves with β-CD is shown in Figure 8f. The scavenging rate of the DPPH radical increased stepwise with increasing FSE and FSE-β-CD concentrations. In addition, the FSE-β-CD EC_50_ (28.98 ug/mL) was lower than FSE (36.61 ug/mL) and 1:1 FSE/β-CD PM (45.08 ug/mL). The scavenging rate of the ABTS radical was the same as that of the DPPH, and the scavenging rate of the ABTS radical also increased stepwise (Figure 8g). The FSE-β-CD EC_50_ was 25.54 ug/mL, which was lower than the FSE EC_50_ of 29.8 ug/mL and 1:1 FSE/β-CD PM EC_50_ of 31.14 ug/mL. The results showed that the antioxidant activity of FSE was improved significantly due to the formation of inclusion complexes of FSE and β-CD. The free radical scavenging ability of a compound was closely related to its hydrogen donating ability [33]. Hydrogen bonds and van der Waals forces were generated between FSE and β-CD, and then the degree of hydroxylation, solubility and stability of guest molecules increased, making it easier to donate hydrogen and react with free radicals such as ABTS and DPPH. In addition, cyclodextrin entrapped extraction, which increased the natural product content of *Forsythia suspensa* leaves in the products, thereby, increasing the antioxidant properties. Some studies have also found that the antioxidant activities of compounds become stronger after complexation, such as cannabidiol [30] and quercetin [34].

## 3. Materials and Methods

### 3.1. Materials and Reagents

The fried green *Forsythia suspensa* leaves used in this experiment were provided by the Hebei Forsythia Industrial Technology Research Institute (Handan, Hebei, China). The raw materials were ground and sieved (40 mesh). β-CD was purchased from Sinopharm Chemical Reagent Co., Ltd. (Shanghai, China). Forsythoside A, phillyrin and phillygenol standards were purchased from Desite Biotechnology Co., Ltd. (Sichuan, China). Other chemical reagents were of analytical grade and were obtained from local commercial sources.

### 3.2. Optimization of Extraction Using RSM

A certain amount of fried green *Forsythia*
*suspensa* leaf powder and β-CD aqueous solution were mixed and stirred with a magnetic stirrer at a certain temperature for 1 h. After that, the mixed solutions were centrifuged at a speed of 4000 r/min for 5 min, and then the supernatant was collected to be stored in centrifugal tubes. Some of the supernatant was used for HPLC analysis, and the rest of the supernatant was sprayed and dried for subsequent experiments. The extraction parameters, including the R*_S/L_*, R*_F/β_*, extraction temperature and pH, were optimized using single-factor experiments and response surface methodology.

In the single-factor experiments, each of these factors was varied at five levels: the solid–liquid ratio (R*_S/L_*) at 1:10, 1:20, 1:30, 1:40 and 1:50, the ratio of β-CD and *Forsythia suspensa* leaves (R*_β/F_*) at 5:1, 5:1, 5:3, 5:4 and 5:5, the extraction temperature at 25, 40, 55, 70 and 85 °C and the pH at 1, 3, 5, 7 and 9. Similarly, the same was true for pure-water extraction except that β-CD was not added.

After the key factors were determined with single-factor experiments, a four-factor RSM method was used to study the relationship between response variables and process variables. The response variables were predicted by the quadratic model, as shown in Equation (2).
(2)$$Y=A+\overset{3}{\underset{i=1}{\sum }}BiXi+\overset{3}{\underset{i=1}{\sum }}\;Cii\;X_{i}^{2}+\overset{2}{\underset{i=1}{\sum }}\overset{3}{\underset{j=i+1}{\sum }}CijXiXj$$
where *Y* was the predicted dependent variable, *X_i_* were the independent variables, *A* was the constant coefficient, *B_i_* were the linear regression coefficients, *C_ij_* were the interaction effect terms and *C_ii_* were the quadratic effect terms, respectively.

Independent variables or process variables (*X*_i_) and their levels were: R*_S/L_* (*X*_1_, 1:20~1:40); pH (*X*_2_, 1~5); extraction temperature (*X*_3_, 55~85 °C); and R*_F/β_* (*X*_4_, 3:5~5:5). All variables were coded at three levels (−1, 0 and 1) according to Equation (3).
(3)$$x_{i}=\frac{X_{i}-X_{0}}{\Delta X_{i}},\;i=1,\;2,\;3$$
where *x_i_* and *X_i_* referred to the dimensionless and actual values of the independent variable i, respectively. ∆*X_i_* represented the step change in *X_i_* and *X*_0_ was the actual value of *i* at the central point.

The response variable was the comprehensive integral (Y), and its calculation method is shown in Equations (4)–(7).
(4)$$Extraction\;yield\;of\;forsythoside\;A\;\left(\%\right)\frac{mass\;of\;extracted\;forsythoside\;A}{mass\;of\;added\;forsythia\;leaves}100\%$$
(5)$$Extraction\;yield\;of\;phillyrin\;\left(\%\right)=\frac{mass\;of\;extracted\;Phillyrin}{mass\;of\;added\;Forsythia\;leaves}100\;\%$$
(6)$$Extraction\;yield\;of\;phillygenol\;\left(\%\right)=\frac{mass\;of\;extracted\;\mathrm{phillygenol}}{mass\;of\;added\;Forsythia\;leaves}100\;\%$$
(7)Y=100 (Extraction yield of Forsythoside A 40%+Extractionyieldofphillyrin 30%+Extraction yield of phillygenol 30 %)

There were 25 experiments with 5 central points, as shown in Appendix A. The least square regression method was used to analyze the experimental data, and the parameters of the mathematical model were obtained. The determination coefficient (R^2^) of the model was obtained using analysis of variance to evaluate the applicability and accuracy of the prediction model. At the same time, the non-significant related terms were excluded from the mathematical equation (*p* > 0.05). In order to explore the influence of various factors on response variables, a three-dimensional response surface diagram was created. Finally, the optimum extraction conditions were determined. 

Pure-water extraction used the three-factor RSM method to study the relationship between response variables and process variables. The independent variables or process variables (X_i_) and their levels were: R*_S/L_* (X_1_, 1:10~1:30); pH (X_2_, 3~7); and extraction temperature (X_3_, 55~85 °C). There were 17 experiments including 5 central points, as shown in Appendix A. Other experimental conditions were the same as those of the cyclodextrin-assisted extraction.

### 3.3. Quantitative Analysis of Forsythoside A, Phillyrin and Phillygenol Using HPLC

HPLC was carried out on an Agilent 1260 system (Agilent Technology, Palo Alto, Santa Clara, CA, USA) to determine the concentration of forsythoside A, phillyrin and phillygenol. The separation process was conducted on a Hypersil GOLD C18 (4.6 × 250 mm, 5 µm) column with a mobile phase containing methanol and 0.2% acetic acid in the ratio: 0−4 min, 32% A; 4~27 min, 32% A→55% A; and 27~35 min, 55% A. The flow rate was 0.5 mL/min. The column temperature was maintained at 30 °C and the detection time was 45 min. The wavelength detected was 229 nm and the volume of each injection was 5 μL. All the samples were filtered through a 0.22 μm membrane filter before being analyzed.

### 3.4. Characterization of Natural Products from Forsythia Suspensa Leaves

#### 3.4.1. Fourier Transform Infrared Spectroscopy (FT-IR)

FSE, β-CD, FSE-β-CD, FSE and β-CD 1:1 physical mixture (1:1 FSE/β-CD PM) were fully mixed with an appropriate amount of KBr and then compressed into KBr disks, respectively. Then, the FT-IR spectrum was obtained using a Nicolet FT-IR spectrometer (Thermo Fisher Scientific, Waltham, MA, USA). Each sample was scanned 64 times between 400 and 4000 cm^−1^ with an optical resolution of 4 cm^−1^.

#### 3.4.2. Powder X-ray Diffraction (PXRD)

The PXRD patterns of samples (FSE, β-CD, FSE-β-CD and 1:1 FSE/β-CD PM) were obtained using a Smart lab X-ray diffractometer (Rigaku Corporation, Tokyo, Japan) using Cu Kα radiation (λ = 1.5406 Å) at 40 kV/40 mA. The scanning process was carried out from 5 to 90° with a step length of 0.02°.

#### 3.4.3. Differential Scanning Calorimetry (DSC)

DSC analysis of samples was performed using a TG/DTA 7300 analyzer (Hitachi, Japan). Each aliquot of approximately 10 mg of the sample was placed in a sealed aluminum crucible and heated from 30 to 400 °C at a rate of 10 °C/min under a nitrogen atmosphere.

#### 3.4.4. Scanning Electron Microscopy (SEM)

The morphology and surface of the samples were observed using a JSM-6700F cold-field-emission scanning electron microscope (JEOL, Tokyo, Japan). A small piece of double-sided adhesive tape was fixed to the conductive plate, and the sample powder was sprayed on the surface of the stub. After sputter coating with gold, the samples were subjected to analysis.

#### 3.4.5. Molecular Docking

In order to investigate the interaction between natural products and β-CD, molecular docking was performed using the AutoDock Tools program (Version 1.5.7, Scripps Institute, La Jolla, CA, USA). To model the natural products/β-CD IC interactions, we adopted the Lamarckian genetic algorithm (LGA). The binding energy was calculated concerning the ranking functions, and the configurations of the clusters were ordered from the lowest to the highest binding energy. The optimal conformation was further analyzed using PyMol (Version1.7.0, Schrodinger LLC., New York, NY, USA).

### 3.5. Thermal Stability

Appropriate FSE and FSE-β-CD powder were dissolved with 20 mL ultra-pure water, respectively. The solution was transferred into a test tube and stored in a 90 °C water bath for 0, 1/2, 1, 2, 4, 6, 8 and 10 h respectively, and then samples were taken for HPLC analysis of forsythoside A, phillyrin and phillygenol content.

### 3.6. Aqueous Solubility

At 25 °C, FSE and FSE-β-CD powder were continuously added to 20 mL of deionized aqueous solution until the solution was supersaturated. After standing for 1.5 h and centrifuging at 4000 r/min for 5 min, the supernatant was taken and analyzed with HPLC for forsythoside A, phillyrin and phillygenol content.

### 3.7. In Vitro Antioxidant Activity

On account of FSE-β-CD and 1:1 FSE/β-CD PM containing β-CD, much less antioxidant active than FSE, the relative concentrations (RCs) needed to be transformed for calculations, and the RC of FSE-β-CDs was converted from the product-to-product ratio of *Forsythia suspensa* leaves. An amount of 3.4 g of *Forsythia*
*suspensa* leaves allowed the production of 2.77 g of FSE-β-CD and 1 g of FSE, respectively. Then, the FSE-β-CD relative concentration and 1:1 FSE/β-CD PM relative concentration were calculated using the following Equations (8) and (9).
(8)RC of FSE−β−CD=Actual concentration of FSE−β−CD/2.77
(9)RC of 1:1 β−CD PM=Actual concentration of 1:1 β−CD PM/2

#### 3.7.1. DPPH Free Radical Scavenging Activity

A 100 mL methanol solution of DPPH (0.2 mM) was prepared. Subsequently, 2 mL of DPPH solution was added to 1 mL of the sample solution of various concentrations, and an additional 2 mL of methanol was added. The mixture was shaken sufficiently and incubated for 30 min. Finally, the Abs at 517 nm was measured using UV/Vis spectroscopy, and Vc was used as a positive control. The scavenging rate of each sample, expressed as R_SD_ (%), was counted using Equation (10). The 50% effective concentration (EC_50_) value of the samples was determined by plotting the R_SD_ (%) vs. concentration of each sample.
(10)$$R_{SD}=(1-\frac{A_{i}-A_{j}}{\;A_{0}})\times 100\%$$
where *A_i_* represented the absorbance of the reaction system (DPPH with the sample), *A_j_* referred to the absorbance of the sample background (methanol with the sample) and *A*_0_ was the absorbance of the negative control (DPPH with methanol).

#### 3.7.2. ABTS Free Radical Scavenging Activity

A 50 mL methanol solution of ABTS (7 mM) was mixed with an 880 μL aqueous solution of potassium persulfate (140 mM). The obtained mixture was kept in the dark for 24 h at room temperature. The prepared ABTS stock solution was diluted approximately 65 times with methanol until the Abs reached 0.70 ± 0.02 at 734 nm. As a result, the ABTS work solution was obtained. Then, 4 mL of ABTS work solution was added into 1 mL of the sample solution of various concentrations. The obtained solution was shaken quickly for 30 s and placed in the dark for another 6 min. Subsequently, the Abs at 734 nm was measured using UV/Vis spectroscopy, and vitamin C (Vc) was used as a positive control. The scavenging rate of each sample, expressed as R_SD_ (%), was calculated with Equation (11). The 50% effective concentration (EC_50_) of the samples was speculated by plotting the R_SD_ (%) vs. concentration of each sample.
R_SD_(%) = (A_0_ − As)/A_o_ × 100% (11)
where A_0_ referred to the absorbance of the negative control (ABTS with methanol) and A_S_ was the absorbance of the reaction system (ABTS with the sample).

### 3.8. Statistical Analysis

All of the analyses were conducted in triplicate, and the results were expressed as the mean ± standard deviation. Response surface methodology (RSM) was designed and analyzed using the Design-Expert V8.0.5 trial software. Microsoft Office 2016 and SPSS 13.0 were used for other statistical analyses.

## 4. Conclusions

The results demonstrated that the optimized green method utilizing the β-cyclodextrin-assisted process was capable of producing a *Forsythia suspensa* leaf extract rich in natural bioactive compounds. Compared with pure-water extraction, the β-CD complex’s form significantly enhanced the yields of forsythoside A (an increase of 14.16%), phillyrin (an increase of 22.47%) and phillygenol (an increase of 52.77%). The interactions between the *Forsythia suspensa* leaf extract and β-CD were verified using PXRD, FT-IR spectroscopy, DSC, SEM and molecular docking, which confirmed the formation of β-CD inclusion complexes with the active ingredients. It was observed that the stability and water solubility of natural active substances of *Forsythia suspensa* leaves were enhanced because of the presence of β-CD. Furthermore, the physicochemical properties and biological activities of Forsythia suspension leaf extract were improved due to the interaction between β-CD and active substances. The novel β-CD-assisted extraction of Forsythia could predictably be a useful extraction technique. It could accelerate further research in the cosmetic and nutritional industries, as well as with other interesting botanical species.

## Figures and Tables

**Figure 1 molecules-27-07055-f001:**
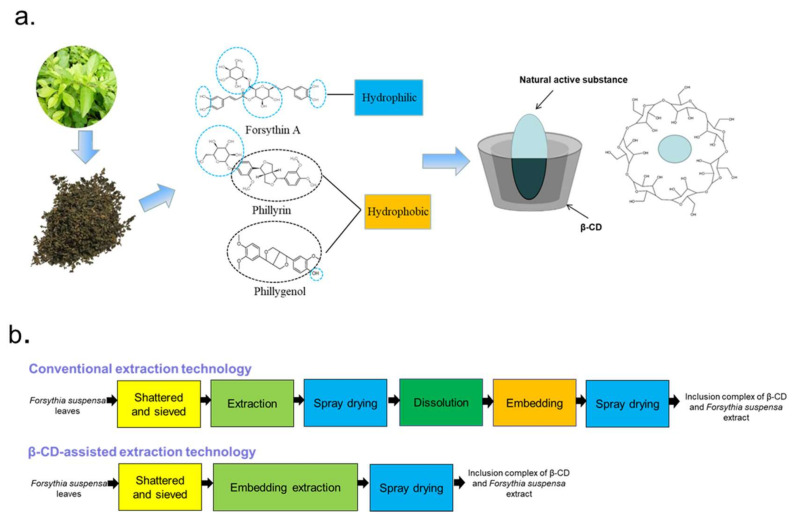
β-cyclodextrin (β-CD)-assisted extraction principle. Natural active molecules’ interaction with β-CD (**a**) and schematic of β-CD-assisted extraction (**b**).

**Figure 2 molecules-27-07055-f002:**
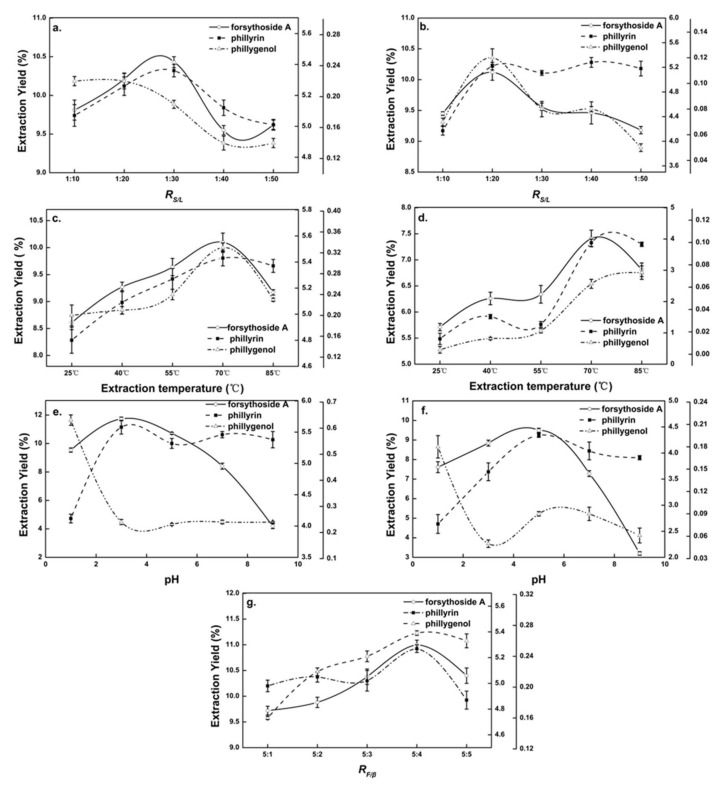
Single-factor diagram of β-CD-assisted extraction and pure-water extraction. Effects of (**a**) solid–liquid ratio (R*_S/L_*) on β-CD-assisted extraction, (**b**) R*_S/L_* on pure-water extraction, (**c**) extraction temperature on β-CD-assisted extraction, (**d**) extraction temperature on pure-water extraction, (**e**) extraction pH on β-CD-assisted extraction, (**f**) extraction pH on pure-water extraction and (**g**) β-CD and *Forsythia suspensa* leaf ratio (R*_F/β_*) on β-CD-assisted extraction.

**Figure 3 molecules-27-07055-f003:**
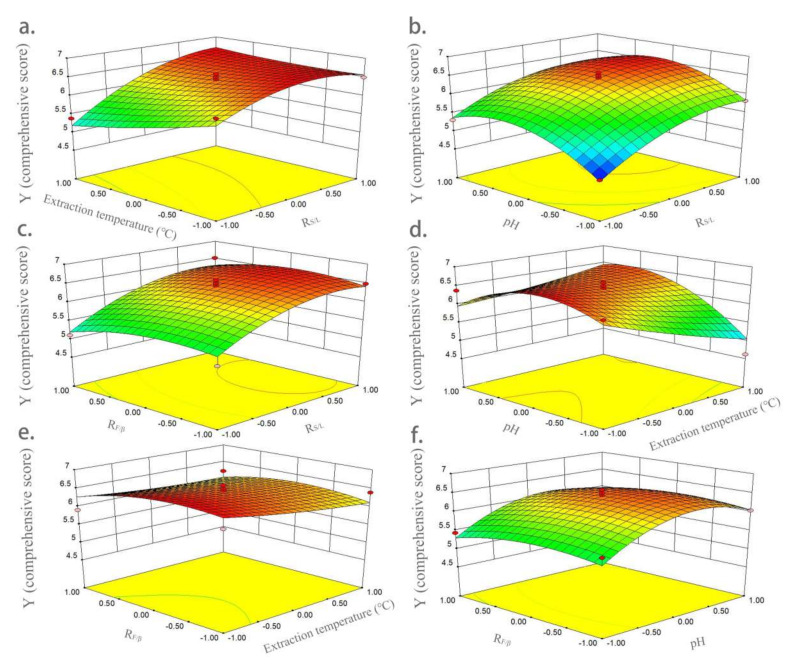
Three-dimensional response surface graphs of the influences of variables on comprehensive score. (**a**) Interaction of extraction temperature and R*_S/L_*; (**b**) interaction of pH and R*_S/L_*; (**c**) interaction of R*_F/β_* and R*_S/L_*; (**d**) interaction of pH and extraction temperature; (**e**) interaction of R*_F/β_* and extraction temperature; and (**f**) interaction of R*_F/β_* and pH.

**Figure 4 molecules-27-07055-f004:**
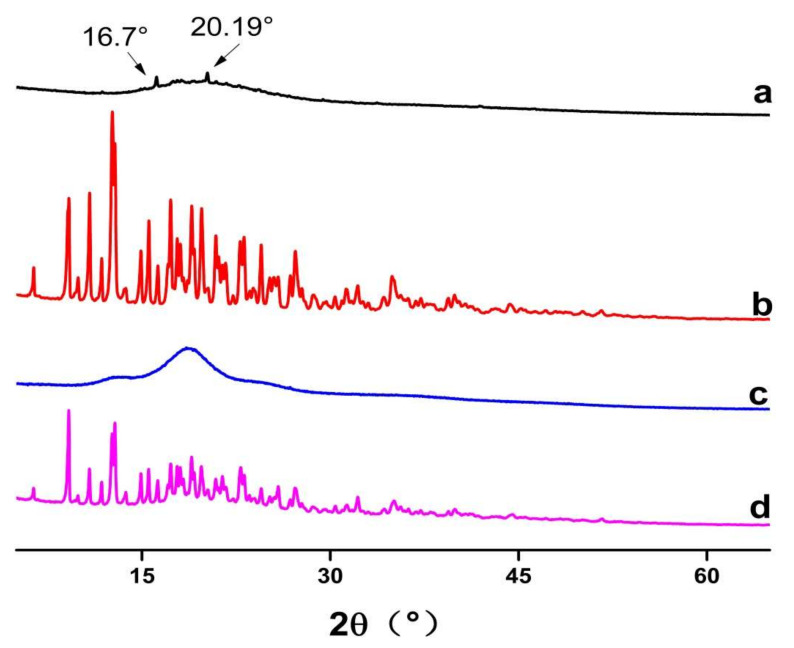
Powder XRD diffractograms of (**a**) *Forsythia suspensa* leaf extract (FSE), (**b**) β-CD, (**c**) β-CD-assisted extract of *Forsythia suspensa* leaves (FSE-β-CD) and (**d**) β-CD with *Forsythia suspensa* leaf extract 1:1 physical mixture (FSE/β-CD PM).

**Figure 5 molecules-27-07055-f005:**
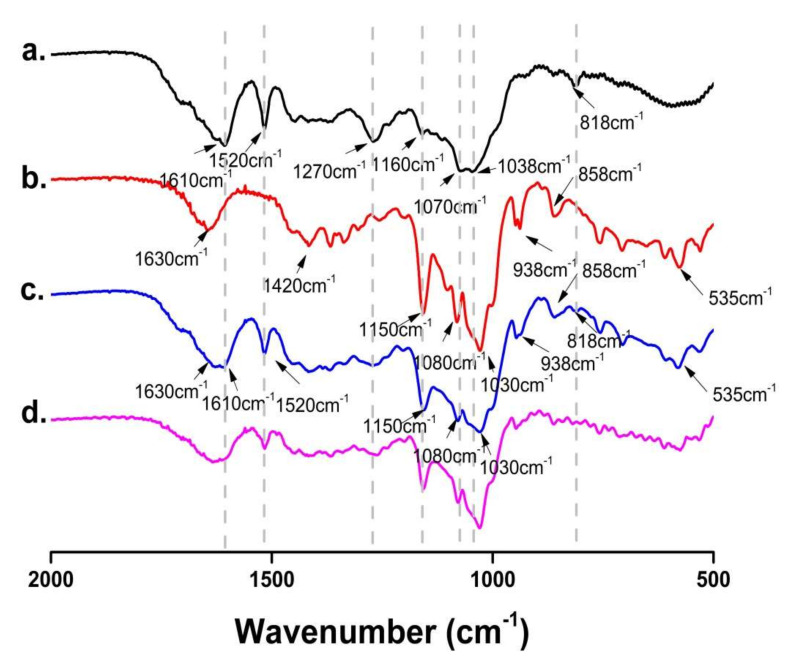
FT-IR spectra of (**a**) *Forsythia suspensa* leaf extract (FSE), (**b**) β-CD, (**c**) β-CD-assisted extract of *Forsythia suspensa* leaves (FSE-β-CD) and (**d**) β-CD with *Forsythia suspensa* leaf extract 1:1 physical mixture (FSE/β-CD PM).

**Figure 6 molecules-27-07055-f006:**
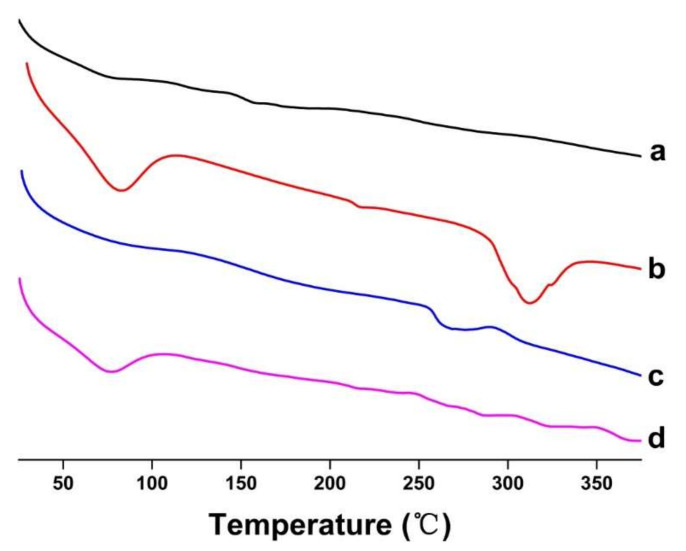
DSC thermograms of (**a**) *Forsythia suspensa* leaf extract (FSE), (**b**) β-CD, (**c**) β-CD-assisted extract of *Forsythia suspensa* leaves (FSE-β-CD) and (**d**) β-CD with *Forsythia suspensa* leaf extract 1:1 physical mixture (FSE/β-CD PM).

**Figure 7 molecules-27-07055-f007:**
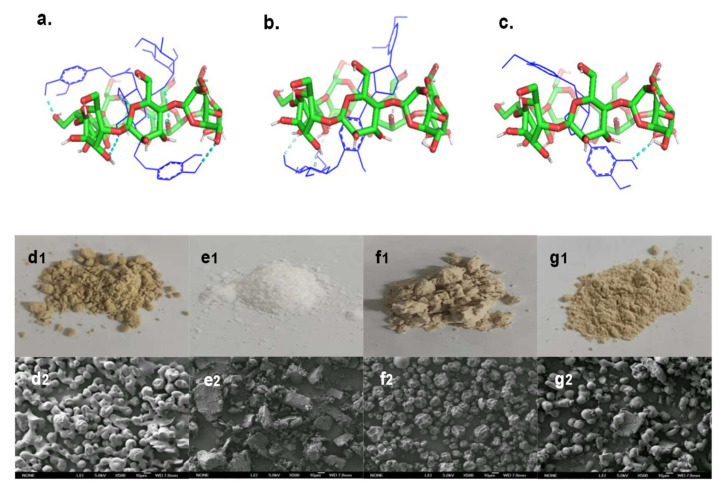
Microstructures of samples and structural optimization and molecular docking of (**a**) forsythoside A/β-CD IC, (**b**) phillyrin/β-CD IC and (**c**) phillygenol/β-CD IC. The sample photos are (**d_1_**) FSE, (**e_1_**) β-CD, (**f_1_**) FSE-β-CD and (**g_1_**) 1:1 FSE/β-CD, and the SEM images of samples are (**d_2_**) FSE, (**e_2_**) β-CD, (**f_2_**) FSE-β-CD and (**g_2_**) 1:1 FSE/β-CD PM.

**Figure 8 molecules-27-07055-f008:**
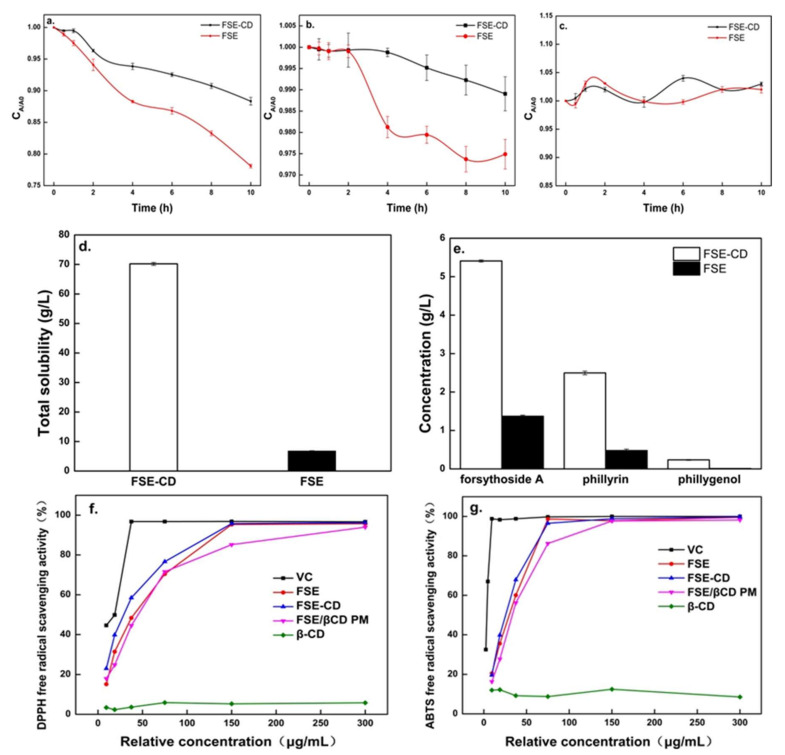
Thermal stability study of extraction of *Forsythia suspensa* leaves, aqueous solubility experiments of FSE-β-CD and FSE and in vitro antioxidant activities of samples. Stability of forsythoside A (**a**), phillyrin (**b**) and phillygenol **(c),** total solubility of FSE-β-CD and FSE (**d**), solubility of natural active substances (**e**), DPPH radical scavenging activity **(f)** and ABTS radical scavenging activity (**g**).

## Data Availability

Not applicable.

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
