# Peer review of "Green Extraction of Forsythoside A, Phillyrin and Phillygenol from Forsythia suspensa Leaves Using a β-Cyclodextrin-Assisted Method"

_molecules, 2022, doi:10.3390/molecules27207055_

Round 1

Reviewer 1 Report

The use of beta-cyclodextrin and other native cyclodextrins to assist in the extraction of active components from plant sources is a method of increasing popularity. For this reason, the present manuscript is an interesting contribution.

There is, however, a very small number of references on this topic. More references should be added to the introduction. Some examples (non-vinvulative, authors can choose other references if they are more pertinent) are given below:

-"β-cyclodextrin enhanced ultrasound-assisted extraction as a green method to recover olive pomace bioactive compounds", doi: 10.1111/jfpp.15194

- "Cyclodextrin-Assisted Extraction Method as a Green Alternative to Increase the Isoflavone Yield from Trifolium pratensis L. Extract", doi: 10.3390/pharmaceutics13050620

- "β-Cyclodextrin-Assisted Extraction of Polyphenols from Vine Shoot Cultivars", doi: 10.1021/acs.jafc.5b00672

In addition, thee are some errors in the scientific nomenclature employed in the manuscript that need to be corrected.

- Cyclodextrins form inclusion compounds. The term "embedding" is inadequate and it needs to be removed from the entire manuscript. When there is insufficient evidence of inclusion compound formation, other expressions can be used such as "interaction with the cyclodextrin"

- Powder diffraction is not a spectroscopic technique, and thus "PXRD spectra" is the wrong term. Replace with "powder XRD diffractogram", or "PXRD pattern".

- There is no such thing as "inclusion peak" in PXRD. Inclusion compounds present a set of reflections that are characteristic of a new crystalline phase formed by the units of the inclusion compound. In the case of the present work, the material is amorphous and NO conclusions can be obtained from this technique. The sentence in lines 205-206 must be removed.

Figures for the different techniques: FTIR, PXRD and DSC should be presented separately and with higher resolution.

- Powder diffractograms can be shown in the scale from 0 to 60 degrees of two-theta as above this value no diffraction is observed.

- FTIR spectra must be shown in the region from 1700 to 500 cm-1 because this region is where the peaks of interest are observed and making the xx axis smaller helps the perception of the different peaks and their possible shifts.

Reviewer 2 Report

The manuscript "Green extraction of forsythoside A, phillyrin and phillygenol 2 from forsythia suspensa leaves using β-cyclodextrin assisted 3 method" deals with the use of a β-CD assisted extraction process of active ingredients from forsythia suspensa leaves. 

The work is innovative and well organized; but some revisions are required before the publication, as follows:

Abstract. Define acronyms the first time they appear.

Introduction. The state of the art on the extraction of active principles from vegetable matter can be enlarged, comparing this method with other innovative technologies, like supercritical fluid extraction. For this purpose, see for instance the works of Baldino and Reverchon, De la Ossa et al., Mainar et al., etc..

Typos have to be corrected. Check journal template.

R&D. The explanation of the obtained results should be enlarged; in particular, the phenomena observed have to be discussed accurately since the only addition of references is not enough.

Conclusions and Abstract are too similar. Rewrite the last paragraph in a more critical way.

Round 2

Reviewer 1 Report

The authors have performed the revisions on the manuscript that were suggested in the first revision round. Some problems are, however, still needing further correction:

1) Powder diffraction. The authors claim, on lines 237-237, that there is a difference in the diffractogram of the physical mixture and the product extravted in the presence of the cyclodextrin. This is not true. The diffractograms have exactly the same diffraction peaks, at the same values of 2theta. The only difference is in the peak intensity which is explained due to the different amount of cyclodextrin in each sample or even due to different amounts of sample used in each diffraction reading. The powder diffraction data presented does not allow to obtain conclusions regarding the presence or not of an inclusion compound, or even regarding the presence or not of interactions between the cyclodextrin and the components of the extract. These conclusion will have to be obtained from other characterisation methods.

2) there is a problem with the infrared spectra (Figure 5). The traces do not look like proper lines, but rather like "stairs". This is usually due to the use of too little decimal figures when exporting the data from the spectrometer to an excell-compatible file (wy, ascii, etc). Please save the data again using four or more decimal figures and replot the spectra.

3) In infrared, it is important to compare the peaks of the pure extract with those of the extract prepared in the presence of cyclodextrin, that is labels should be on traces (a) and (c). Please label the main peaks in the spectrum (C) and comment on any important shits. The band observed at 1610 in the pure extract is an important probe for possible interactions with cyclodextrin, if one can observe a shift in FTIR this will be most helpful. But we need proper resolution in the spectra for that to be observed (see point2)

4) The descriptio of the thermal phenomana occuring in the DSC trace of the physical mixture is very confusing. Please rephrase it or remove it, because it does not seem to contribute in any way to the understanding ot the processes happening in the cyclodextrin-assisted extract.

5) Some moderation is needed in the conclusions section, in particular for the text in lines 544-547. The fact that the cyclodextrin-assisted method worked well with the plant studied in the present work does not imply that it will be sucessful with all the other plants. Please re-write this part in order to reflect such fact. Maybe something like 'the present work shows the method is useful for forsythia and it encourages further research with other botanical species of interese to the cosmetic and nutraceutical industries'.

Reviewer 2 Report

The authors performed the modifications suggested by the Reviewer; however, the refs added do not correspond to the text. For supercritical fluid technology, take a look, for instance, at these works: Baldino et al., Journal of Supercritical Fluids, 2018, 131, 82–86; Kaul et al., Molecules, 2022, 27, 80; etc...
